# Anomalous universal conductance as a hallmark of non-locality in a Majorana-hosted superconducting island

Yiru Hao[1,2,4], Gu Zhang[3,4], Donghao Liu[1] & Dong E. Liu [1,2,3] ✉

The non-local feature of topological states of matter is the key for the topological protection of quantum information and enables robust non-local manipulation in quantum information. Here we propose to manifest the non-local feature of a Majorana-hosted superconducting island by measuring the temperature dependence of Coulomb blockade peak conductance in different regimes. In the low-temperature regime, we discover a coherent double Majorana-assisted teleportation (MT) process, where any independent tunneling process always involves two coherent non-local MTs; and we also find an anomalous universal scaling behavior, i.e., a crossover from a $[\max(T,eV)]^6$ power-law to a $[\max(T,eV)]^3$ power-law conductance behavior when energy scale increases – in stark contrast to the usual exponential suppression due to certain local transport. In the high-temperature regime, the conductance is instead proportional to the temperature inverse, indicating a non-monotonic temperature-dependence of the conductance. Both the anomalous power law and non-monotonic temperature-dependence of the conductance can be distinguished from the conductance peak in the traditional Coulomb block, and therefore, together serve as a hallmark for the non-local feature in the Majorana-hosted superconducting island.

The topological states of matter provide a non-local way to store quantum information via their degenerate topological ground states[1]. This non-local nature is the key for the topological protection of quantum information[2], and, more importantly, enables non-local manipulation in quantum information, via e.g., fusion and braiding[1–3] of hosted non-abelian anyons. The hotly debated Majorana zero modes (MZMs)[4,5], which recently attracts a lot of experimental activities[6–25], is still among the simplest and most promising candidates of non-abelian anyons. The proposed fusion and braiding tests[2,26–28] however require sophisticated experimental devices and procedures beyond the scope of current technology. It is thus rewarding to first reveal the non-local feature via better-developed techniques, including the quantum transport.

Electron transport through confined quantum islands is usually influenced by electrostatic energy, leading to the Coulomb blockade (CB) signatures with conductance oscillations [29]. Relying on non-local electrostatic interactions, the CB effect offers a natural playground in the detection of non-locality. In the presence of superconductivity (SC), the signature of CB is modified. When the order parameter is larger than the charging energy, the single electron (or $1e$) tunneling is suppressed and only the $2e$ Cooper pair tunneling survives, leading to the oscillation with $2e$ periodicity[30]. This $2e$-feature maybe however not the case when facing a topological SC island. Indeed, the non-local transport through a topological SC island[31,32], known as the topological Majorana-assisted teleportation (MT), can be generated from topological degeneracy and the Coulomb charging energy[32], and gives a $1e$ periodicity in CB. Afterward, a more careful theoretical analysis was carried out to obtain the CB signatures[32–35], including: (1) for all different cases, the CB peak height increases while lowering the

[1]State Key Laboratory of Low Dimensional Quantum Physics, Department of Physics, Tsinghua University, 100084 Beijing, China. [2]Frontier Science Center for Quantum Information, 100184 Beijing, China. [3]Beijing Academy of Quantum Information Sciences, 100193 Beijing, China. [4]These authors contributed equally: Yiru Hao, Gu Zhang. ✉e-mail: dongeliu@mail.tsinghua.edu.cn

temperature; (2) CB oscillations with 1$e$ and 2$e$ period, respectively, accompany the tunneling of 1$e$ quasiparticles and 2$e$ Cooper pairs, and (3) The CB peak shape of MT is the same as that of a resonant level model[32,33] captured by Breit–Wigner formula[36,37]. Because of these coincidences with the standard CB features, it is not yet known whether or not the two-terminal CB island could provide a hallmark for non-local features of a Majorana-hosted (either topological Majorana or quasi-Majorana) SC island.

In this work, we study the two-terminal transport through a CB island that hosts a MZM (or a quasi-Majorana pair) and two coupled MZMs (with the coupling amplitude $v$) at opposite sides of the island (Fig. 1a). Based on our analysis, when $v$ is much larger than the coupling $\Gamma_R$ between the right lead and the uncoupled MZM, such a Majorana-hosted SC island displays unique transport features. As the starter, the 1$e$ conductance peak locations are independent of the value of $v$. This is in stark contrast to the MT where the peak position is inter-MZM coupling dependent[33]. More interestingly, our system is expected to display a non-monotonic temperature dependence at the 1$e$ CB conductance peak (Fig. 1b). In the lowest energy regime, unlike the usual exponential conductance behavior through localized high-energy states, we predict an exotic power-law scaling $\sim[\max(T,eV)]^6$ for the peak conductance due to a novel non-local coherent double MT, where any tunneling event connecting two leads involves two coherent MT processes. When energy increases (above the level broadening), the paired MTs lose coherence and the conductance crosses over to another power-law $\sim[\max(T,eV)]^3$ scaling. Further increasing the energy, the 1$e$ peak height reaches its maximum when the energy is around the inter-MZM coupling $v$. Above this energy, the 1$e$ peak height starts to decrease and approaches the standard MT results[33] with the -1/$T$ scaling. The anomalous conductance features, which rely greatly on the non-locality of the coherent double MT, can be used as hallmarks for Majorana-assisted non-local transport, as they are in sharp contrast to those of normal CB systems. We emphasize that our results apply to systems with either topological Majorana or a quasi-Majorana pair. Of the latter case, only one of these two non-communicating quasi-Majoranas participates the non-local transport. Meanwhile, the double Majorana-assisted teleportation is enforced to occur due to the finite ABS energy $v$: this is different from the double zero-energy ABS structure considered in ref. 38.

## Results

### Model and low-energy effective theory

One possible realization of the proposed system is shown in Fig. 1c, where a floated superconductor-proximitized nanowire (the pink line) weakly couples to one normal lead at each side. Under the protection of the smooth potential[39], two pairs of partially separated MZMs (or quasi-Majoranas) emerge at two ends of the nanowire in the (topologically) trivial regime[39–48]. As shown in Fig. 1c, we can model our system with four quasi-Majoranas at each end as $\gamma_1$, $\gamma_2$, $\gamma_3$, and $\gamma_4$. With them, we construct two independent auxiliary fermionic operators $d_1 = (\gamma_1 + i\gamma_4)/2$ and $d_2 = (\gamma_2 + i\gamma_3)/2$. We tune the left tunneling barrier into a relatively steep shape to partially overlap $\gamma_1$ and $\gamma_2$ with the coupling strength $v$, and consider the regime that only $\gamma_1$ of the pair effectively coupled to the left lead. The $\gamma_2$-lead coupling is exponentially suppressed and thus neglected. In addition, we keep the right barrier in a shallow shape to make sure the coupling between the other pair is negligible[47]. Of this structure, transport through the island is dominated by MT, where the MZM $\gamma_4$ plays an essential and unique role. The change of the status of $\gamma_4$ is then expected to generate a manifest (qualitative) modification of the conductance feature. This is in sharp contrast to transport with sequential local tunnelings. Indeed, of the latter case, sites of the superconducting island are almost effectively equivalent, and the hybridization of a single site induces only a minor (quantitative) modification on conductance (Fig. 1d).

For the proposed Majorana-hosted island system (shown in Fig. 1a), the total Hamiltonian can be written as

$$H = H_{\text{lead}} + U_c + H_{\text{coupling}} + H_{\text{T}}, \qquad (1)$$

**(a)**

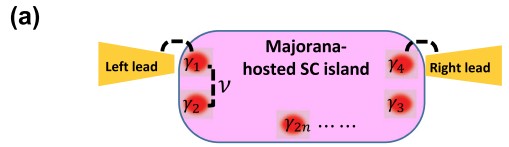

**(b)**

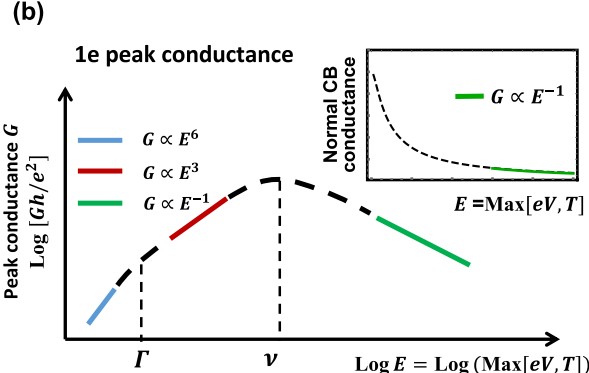

**(c)** Four Quasi-Majorana modes in topological trivial regime

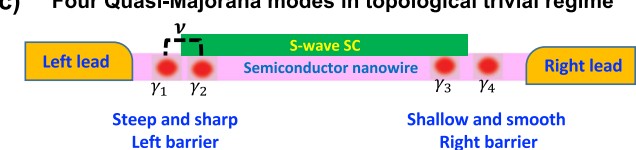

**(d)**

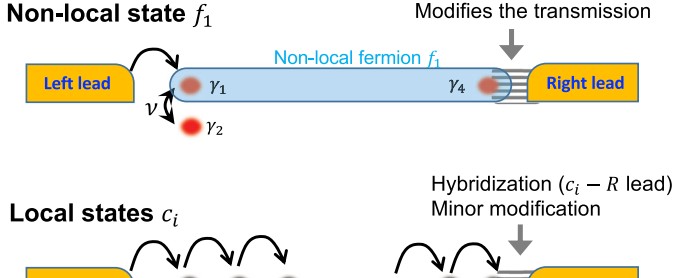

**Fig. 1 | Possible system setups and our major predictions. a** Illustration of our system that consists of four or more MZMs on opposite sides of the nanowire. Two of them ($\gamma_1$ and $\gamma_2$) are coupled with the strength $v$, and only $\gamma_1$ and $\gamma_4$ are connected to leads. **b** The CB peak conductance through our Majorana-hosted island that is tuned to the half-filling. For more intuitiveness, we use logarithmic coordinates and find that it is non-monotonic in temperature. Inset: the monotonic temperature-dependence peak conductance of normal CB peaks. **c** Illustration of another possible realization of our system. **d** A picture on what happens after the hybridization of $f_1$. Upper panel: for non-local transport, the hybridization of $f_1$ changes the status of MZMs, including $\gamma_4$ (which is essential for "teleportation") near the right lead, and also $\gamma_1$, $\gamma_2$ near the left lead. The latter two are also influenced by a finite coupling $v$, which is another element that both manifests non-locality and also distinguishes our work from the MT of ref. 32. This structure leads to a great modification of the tunneling across the island. Lower panel: for local transport, the hybridization of the site next to the right lead simply reduces the effective size of the island, leading to a minor modification of the tunneling.

where $H_{lead} = \sum_{k,j=L,R} \epsilon_j(k) c_{j,k}^\dagger c_{j,k}$ describes two non-interacting leads. $U_c = E_c(N - n_g)^2$ is the electrostatic energy induced by the Coulomb interaction between electrons in the nanowire island. $E_c$ is the charging energy which is smaller than the proximity SC gap but larger than other relevant energy scales. $N$ represents the total number of electrons in the island, and $n_g$ is tunable through a backgate voltage. $H_{coupling} = i v \gamma_1 \gamma_2$ is the coupling term between $\gamma_1$ and $\gamma_2$. As $\gamma_1$ and $\gamma_2$ are both close to the left lead (Fig. 1a), their coupling $v$ does not change the conductance peak position (i.e., $n_g = 2n_0 + 1/2$, where $n_0$ indicates the number of hosted Cooper pairs). This is in stark contrast to the $v$-dependent peak position of a MT, where the inter-MZM coupling is between two non-local MZMs through which the non-local transport is realized. Neglecting the contribution of the quasi-particle states above the SC gap to the electric current at low energies, the tunneling Hamiltonian is

$$H_T = \lambda_L \sum_{k,L} c_{kL}^\dagger \gamma_1 e^{-i\varphi/2} + \lambda_R \sum_{k,R} c_{kR}^\dagger \gamma_4 e^{-i\varphi/2} + h.c., \quad (2)$$

where $\lambda_{L,R}$ denotes the respective tunnel matrix elements, and $e^{\pm i\varphi/2}$ raises/lowers $N$ by one charge unit[49].

Due to the Coulomb blockade, we can further map the model to its low-energy sector. With $n_g$ a half-integer ($n_g = 2n_0 + 1/2$), we only need to consider states in the Hilbert space $\{|00\rangle, |10\rangle, |11\rangle, |01\rangle\}$ spanned by basis vectors that dominate low-energy current tunneling, where $|i,j\rangle$ refers to the state with particle numbers $i$ and $j$ respectively for $d_1$ and $d_2$. To further explore the relevance to MT[32], we define two impurity operators: one fermionic $f_1 = |00\rangle\langle10| - |11\rangle\langle01| = (d_1 - d_1^\dagger)\exp(-i\varphi/2)$ and one bosonic $f_2 = |00\rangle\langle11| - |10\rangle\langle01| = -d_1 d_2 - d_1^\dagger d_2^\dagger$. They are independent since $[f_1, f_2] = \{f_1, f_2\} = 0$. The bosonic operator $f_2$ is equivalent to a spin operator, via the mapping $f_2 = S_-, f_2^\dagger = S_+$, and $S_z = f_2^\dagger f_2 - 1/2$ (see Supplemental Material for more details.). With analysis above, for the peak positions (i.e., half-filling $n_g = 1/2$), the effective Hamiltonian becomes

$$H_{eff} = H_{leads} - 2vS_y - 2\lambda_L \sum_k c_{kL}^\dagger S_z f_1 + \lambda_R \sum_k c_{kR}^\dagger f_1 + h.c., \quad (3)$$

where we have used the fact that $S_y = i(-S_+ + S_-)/2$. In contrast to pioneering topological Kondo setups[49–52] tuned to the Coulomb valley (more precisely, the Kondo valley with $n_g$ an odd integer), transport features predicted by us emerge at energy-degenerate (half-filling) points. More recently, ref. 53 studies the multichannel charge Kondo effect where the island is also tuned to half-filling. Nevertheless, all these topological Kondo setups, as far as we know, require at least three terminals. By contrast, only two terminals are needed in our setup, lowering the difficulty of possible experimental measurements.

It is instructive to study the equilibrium conductance behavior in Eq. (3) at zero temperature. The impurity Hamiltonian $-2vS_y$ has its ground state $|G\rangle = (-i, 1)^T$ which has a zero $S_z$ expectation $\langle G|S_z|G\rangle = 0$. Consequently, the island tunneling to the left lead vanishes at zero-energies ($T = eV = 0$), leading to a zero conductance at the low-energy fixed point. This result can be understood that the influence of the coupling term is to form a localized Andreev-bound state that prevents non-local tunneling completely at zero energies.

## Double Majorana-assisted teleportation at low energy

Let us first analyze the fluctuations near the low-energy fixed point of the effective Hamiltonian Eq. (3) using the leading irrelevant operator. Eq. (3) tells us that the tunneling at the left lead $\propto \lambda_L$ changes the impurity between the low-energy and the high-energy states. This is classically forbidden when $v \gg \max(T, eV), \Gamma_R$, as the energy of the high-energy state is unaffordable by either thermal ($-T$), quantum ($\sim \Gamma_R = \pi\rho|\lambda_R^2|$, where $\rho$ refers to the lead density of states) fluctuations, or the non-equilibrium driving ($-eV$). Quantum mechanically, however, tunneling is possible via high-order tunneling operators that transport

particles through high-energy virtual states. More specifically, when $f_1$ is occupied, we can construct a higher-order tunneling operator with three sub-operators: (i) $c_{qL}^\dagger S_z f_1$, (ii) $f_1^\dagger c_{kR}$ and (iii) $c_{pL}^\dagger f_1 S_z$. Each operator alone is forbidden at low energies due to the energy penalty [(i) and (iii)] or Pauli exclusion principle [(ii)]. However, if high-energy states occur virtually, these operators together combine into a higher-order operator $c_{pL}^\dagger S_z f_1 \cdot f_1^\dagger c_{kR} \cdot c_{qL}^\dagger S_z f_1$ (labeled as process A) that bridges two energy-allowed real states. To produce a persistent current, process A is followed by the operator $c_{kR} f_1^\dagger$ that returns the island to its initial state (labeled as process B). The successive occurrence of processes A and B leads to a persistent electron transport from the right to the left lead. Noteworthily, one needs a careful treatment of the operator $\mathcal{O}_A$ of process A, since it involves two fermionic operators in the left lead. Indeed, after a careful Schrieffer-Wolff transformation (see "Methods", or ref. 54), the process A operator

$$\mathcal{O}_A = \sum_{\epsilon_p > \epsilon_q, k} \frac{2(\epsilon_p - \epsilon_q)}{v^3} \lambda_L^2 \lambda_R c_{pL}^\dagger S_z f_1 \cdot f_1^\dagger c_{kR} \cdot c_{qL}^\dagger S_z f_1 \quad (4)$$

contains a momentum-dependent prefactor, and a conditional summation $\epsilon_p > \epsilon_q$, where $\epsilon_p$ and $\epsilon_q$ refer to the energy of particles with momenta $p$ and $q$, respectively, of an equilibrium reservoir. This prefactor vanishes in zero-energy (i.e., zero-temperature and in-equilibrium) situations where $\epsilon_p = \epsilon_q = 0$ are both fixed at the Fermi level. For finite-energy situations, $(\epsilon_p - \epsilon_q)^2 \sim [\max(T, eV)]^2$ after the summation over momenta.

In this low-energy situation, the effective transmission rate[55] becomes $\tau_{seq} \equiv 1/\Gamma_{seq} = 1/\Gamma_L^{eff} + 1/\Gamma_R$. where $\Gamma_L^{eff}$ refers to the effective level broadening from the higher-order operator $\mathcal{O}_A$. As a higher-order process, $\Gamma_L^{eff}$ is much smaller than $\Gamma_R$, i.e., the level broadening at the right side. In addition, $\Gamma_L^{eff}$ becomes less important at low energies, since its corresponding operator $\mathcal{O}_A$ is irrelevant in the perspective of renormalization group (RG) analysis. Indeed, following a standard RG analysis[56], a free lead fermionic operator at the boundary produces a scaling dimension 1/2. A simple counting shows that $\mathcal{O}_A$ has a tree-level scaling dimension $3/2 > 1$. It is thus RG-irrelevant and its amplitude $\Gamma_L^{eff}$ becomes increasingly unimportant at low energies, in comparison to the level broadening $\Gamma_R$ of the RG-relevant right-lead coupling. With low-enough energies, $\Gamma_R \gg \Gamma_L^{eff}$, and the sequential tunneling rate

$$\tau_{seq} \sim 1/\Gamma_L^{eff} + 1/\Gamma_R \approx 1/\Gamma_L^{eff} \quad (5)$$

which is almost determined by the effective level broadening $\Gamma_L^{eff}$. With this knowledge in mind, we begin to analyze the system's low-energy conductance features in two limiting cases.

In the extremely low-energy regime $\max(T, eV) \ll \Gamma_R \ll v$, processes A and B are coherent, leading to a coherent double MT as shown in Fig. 2a. Indeed, in this regime, an A or B process alone is forbidden as they relax the $f_1$-right lead hybridization, leading to an energy penalty $\Gamma_R$ that is unaffordable by the fluctuation $\max(T, eV) \ll \Gamma_R$. Consequently, A and B processes always occur coherently. This coherent double MT can be captured experimentally via the low-energy current measurement. Indeed, the impurity operator $f_1$ becomes dynamical in this regime, and $\mathcal{O}_A$ of Eq. (4) now effectively consists of six non-interacting lead fermions. Similar treatment on the hybridization of impurity site has been taken in, e.g., for Kondo[57] and two-impurity Kondo[58] systems. Operator $\mathcal{O}_A$ then has the scaling dimension $\alpha = 6 \times 1/2 = 3$ at low energies, which equals six times that of one free fermionic operator (i.e., $1/2$[59]). This scaling dimension indicates the suppressed tunneling $\sim [\max(T, eV)]^{2(\alpha-1)} = [\max(T, eV)]^4$ at low energies[55,60]. This fact, in combination with the extra power from the prefactor of $\mathcal{O}_A$ in Eq. (4), leads to the expected low-energy conductance $G \propto [\max(T, eV)]^6$, which is anomalous and highly distinguishable from conductance features through normal structures. This high power law in energy is a strong signature of non-local coherent tunneling. Indeed,

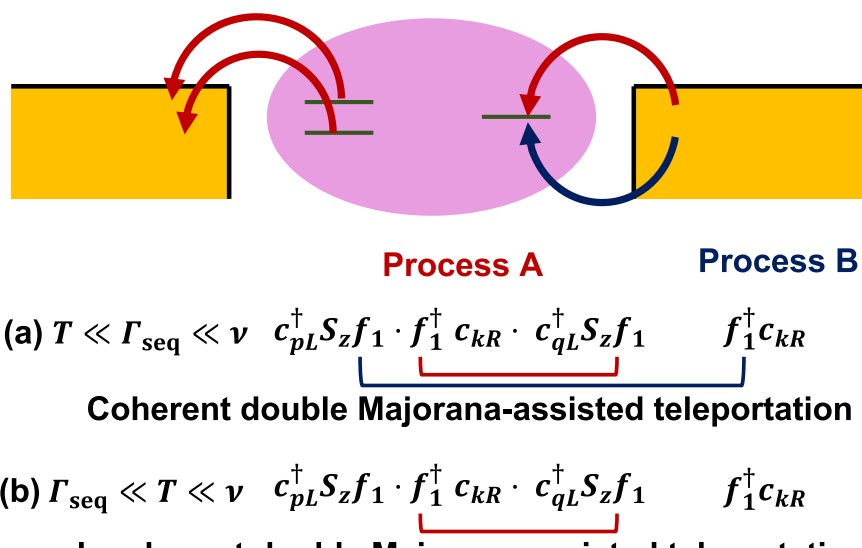

**(a)** $T \ll \Gamma_{\text{seq}} \ll \nu$   $c_{pL}^\dagger S_z f_1 \cdot f_1^\dagger c_{kR} \cdot c_{qL}^\dagger S_z f_1 \qquad f_1^\dagger c_{kR}$

**Coherent double Majorana-assisted teleportation**

**(b)** $\Gamma_{\text{seq}} \ll T \ll \nu$   $c_{pL}^\dagger S_z f_1 \cdot f_1^\dagger c_{kR} \cdot c_{qL}^\dagger S_z f_1 \qquad f_1^\dagger c_{kR}$

**Incoherent double Majorana-assisted teleportation**

**Fig. 2 | The schematic diagrams of high-order coherent operators. a** Process A (red arrow) and B (blue arrow) are coherent in the extremely low-temperature regime $T \ll \Gamma_{\text{seq}} \ll \nu$, where they together construct the coherent double Majorana-assisted teleportation. **b** In the regime $\Gamma_{\text{seq}} \ll T \ll \nu$, these two processes become incoherent.

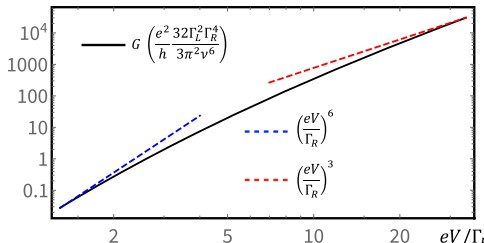

**Fig. 3 | Conductance calculated with Eq. (6) for our system.** The blue and red dashed lines highlight power laws in different limits. The conductance $G \ll e^2/h$ is required in both limits.

the energy-forbidding of local tunneling operators reveals the higher-order non-local events that manifest the deep inner structures of the system.

This anomalous conductance feature accompanies the crossover to another feature for the regime with a slightly higher energy $\Gamma_R \ll \max(T, eV) \ll \nu$. In this regime, the lead-$f_1$ hybridization is relaxed (after which $f_1$ loses its dynamics), thus allowing the individual occurrences of A and B (Fig. 2b). Now, the operator $\mathcal{O}_A$ has the scaling dimension $\alpha = 3/2$ (three times that of a free fermion $c_k$), indicating the low-energy power law $\sim \max(T, eV)$[60]. Once again, we combine this power law with that from the prefactor of $\mathcal{O}_A$, leading to the conductance feature $G \sim [\max(T, eV)]^3$ for low energies.

These two anomalous conductance power laws are among the central points of our work. Briefly, we anticipate the crossover between these power laws in the low-energy regime $\max(T, eV) \ll \nu$: When $\max(T, eV) \gg \Gamma_R$, the conductance is determined by operator $\mathcal{O}_A$, with $G \sim [\max(T, eV)]^3$; When energy decreases, $\mathcal{O}_A$ is modified by the impurity-right lead coupling, and the related conductance feature crosses over to another power law $G \sim [\max(T, eV)]^6$ when finally $\max(T, V) \ll \Gamma_R$. Both the anomalous power laws and the crossover over between them are highly exceptional, and thus capable in the experimental identification of the non-local teleportation.

For clarification, we are aware that polynomial transport features also occur in systems with local transport, e.g., the two-channel Kondo model[61] or a dynamical Coulomb blockade[62] system. However, we

would like to emphasize that in these systems, two metallic leads are separated by a single local impurity. By contrast, in our system, these two leads are instead well-separated by a (finite-width) super-conductor that has a gapped spectrum. In such a system, transmission that relies on local transport will be exponentially suppressed, in contrast to our predicted polynomial features. In addition, we emphasize that the crossover of the global conductance between two different polynomial features occurs after the hybridization of a single fermionic state $f_1$ (consisting of two MZMs). This non-trivial feature also signifies the non-local transport: otherwise the hybridization of a local single site induces only a minor (quantitative) modification of the transport (Fig. 1d). To support our analysis, we calculate the low-bias conductance of our system at zero temperature using Green function technique (see "Methods"). During our calculation, we treat the effective Hamiltonian exactly, except for $\mathcal{O}_A$. Indeed, as $\mathcal{O}_A$ is RG-irrelevant, it is safe to treat $\mathcal{O}_A$ perturbatively to the leading order, where the current becomes

$$I = \frac{2e^2}{h} \int_{-\infty}^{\infty} dt\, e^{ieVt/h} \langle [\mathcal{O}_A^\dagger(t), \mathcal{O}_A(0)] \rangle, \qquad (6)$$

with correlations evaluated without $\mathcal{O}_A$ in the Hamiltonian. In Eq. (6), we have taken the trick (see, e.g., refs. 60, 63 and the "Methods") to deal with the bias as a time-dependent phase factor: by doing so, the correlation can be evaluated as if the system was in equilibrium. The current calculation is tedious but rather straightforward, with which we obtain the exact curve (see "Methods" for the analytical expression) shown in Fig. 3. For two limiting cases, we can show that the conductance yields

$$G \approx \frac{e^2}{h} \frac{4\Gamma_L^2}{45\pi^2\nu^6\Gamma_R^2}(eV)^6 \propto (eV)^6, \quad \text{when } eV \ll \Gamma_R, \qquad (7)$$

$$G \approx \frac{e^2}{h} \frac{16\Gamma_L^2\Gamma_R}{3\pi\nu^6}(eV)^3 \propto (eV)^3, \quad \text{when } eV \gg \Gamma_R. \qquad (8)$$

These low-bias conductance power laws, valid in the regime $eV, \Gamma_R \ll \nu$, perfectly agree with our RG analysis above. Notably, the full conductance range displayed in Fig. 3 is experimentally accessible.

Indeed, the conductance in the $eV \gg \Gamma_R$ limit has the order ~$10^{-2}e^2/h$ (by choosing e.g., $\nu/\Gamma_L = 3$ and $\nu/\Gamma_R = 25$). Around this amplitude, conductance and its corresponding power-law have been shown as experimentally accessible in e.g., refs. 62, 64. In the opposite limit $eV \ll \Gamma_R$, the conductance instead drops to ~$10^{-7}e^2/h$. This high precision, fortunately, has been reported in e.g., ref. 65 when measuring the conductance quanta. In the low-conductance limit, this requirement on the measurement precision is also achieved in e.g., refs. 66, 67. Especially, ref. 66 reports an extremely small conductance ~$10^{-11}e^2/h$, with a visible power-law feature in a long InSb nanowire.

### Single-electron tunneling in the high-temperature regime

In the high-temperature regime $\nu \ll T \ll E_c$, thermal fluctuation allows transport processes (e.g., $c_{pL}^\dagger S_z f_1$) that are otherwise forbidden in low-energies regimes. It is then legitimate to evaluate the conductance via the master equation formalism[29,33–35]. Of our case, the superconducting island contains four eigenstates, $|o_{1,2}\rangle = (\pm i|10\rangle + |01\rangle)/2$ and $|e_{1,2}\rangle = (\pm i|00\rangle + |11\rangle)/2$, where $e$ and $o$ respectively label impurity states with even and odd parities (see Supplemental Material for more details.). The occupation probability of each state follows the rate equations

$$\dot{P}_\alpha = -\sum_\beta \Gamma_{\alpha \to \beta} P_\alpha + \sum_\beta \Gamma_{\beta \to \alpha} P_\beta,$$
$$\dot{P}_\beta = -\sum_\alpha \Gamma_{\beta \to \alpha} P_\beta + \sum_\alpha \Gamma_{\alpha \to \beta} P_\alpha, \tag{9}$$

where $P_\alpha$ and $P_\beta$ are the occupation probability of even $\alpha = |e_1\rangle, |e_2\rangle$ and odd $\beta = |o_1\rangle, |o_2\rangle$ parity states, respectively, and $\Gamma_{i \to f} = \Gamma_{i \to f}^L + \Gamma_{i \to f}^R = \sum_j \Gamma_{i \to f}^j$ represents the transition probability from state $|i\rangle$ to $|f\rangle$. They can be evaluated from the Fermi golden rule

$$\Gamma_{\alpha \to \beta}^j = \frac{2\Gamma_j}{\hbar} \sum_p \delta\left(E_\alpha - E_\beta + \xi_p\right) f\left(\xi_p - \mu_j\right),$$
$$\Gamma_{\beta \to \alpha}^j = \frac{2\Gamma_j}{\hbar} \sum_p \delta\left(E_\beta - E_\alpha - \xi_p\right)\left[1 - f\left(\xi_p - \mu_j\right)\right], \tag{10}$$

where chemical potentials $\mu_L = eV$, $\mu_R = 0$, and $f(\epsilon)$ is the fermionic distribution. $E_\beta - E_\alpha$ is the energy difference between the odd $\beta$ and even $\alpha$ parity states, and $\xi_p$ is the electron energy in the leads.

One can solve Eq. (9) with the normalization requirement $\sum_\alpha P_\alpha + \sum_\beta P_\beta = 1$. With them, the current can be evaluated via $I = e\sum_{\alpha,\beta} P_\alpha \Gamma_{\alpha \to \beta}^L - e\sum_{\alpha,\beta} P_\beta \Gamma_{\beta \to \alpha}^L$. At zero bias, the tunneling conductance becomes (see Supplemental Material for more details)

$$G = \frac{e^2}{2T\hbar} \frac{\Gamma_L \Gamma_R}{\Gamma_L + \Gamma_R} \operatorname{sech}\left(\frac{\nu}{T}\right)^2 \operatorname{sech}\left[\frac{E_c(1 - 2\delta_g)}{2T}\right]^2. \tag{11}$$

In agreement with our previous analysis, the conductance arrives at its peak value at half-filling $\delta_g = n_g - 2n_0 = 1/2$, independent of the inter-MZM coupling $\nu$. As another feature, the peak conductance follows ~$1/T$ in the high-temperature $\nu \ll T \ll E_c$ limit, where the factor $\operatorname{sech}(\nu/T)$ approximately equals one. In the above calculation, the equilibration is reached from the self-consistent treatment of only the lead-island couplings. However, if the thermal effects of the island is mainly from the external environment, the island will first reach the thermal equilibrium. We call this situation "dirty" transport, and the conductance formula becomes slightly different (see Supplemental Material for more details).

Combining the analysis in the low-energy regimes ($\max(T,eV) \ll \nu$) and the rate-equation calculations in the high-energy regime ($\nu \ll \max(T,eV) \ll E_c$), we obtain the 1$e$ conductance-peak features over the main energy regimes, as shown in Fig. 1b. Here, the energy that induces the largest conductance is expected to be around $\max(T,eV) \sim \nu$, as given by the rate-equation result Eq. (11). Indeed, the semi-classical rate-equation is legitimate near this regime, where charge transport mainly relies on uncorrelated sequential tunnelings. In the low-energy limit, conductance predicted by Eq. (11) decays exponentially, instead of the polynomial feature predicted for coherent tunneling operators. In this limit, one needs to go beyond the semi-classical picture, as coherent tunneling has become dominant.

## Discussion

We mostly focus on the 1$e$ CB conductance peak, i.e., $\delta_g = 1/2$, of our Majorana-hosted SC island. We discover a novel double MT and anomalous Coulomb blockade, which manifest the deep inner structures of the system and could serve as a hallmark for the non-local transport in Majorana-hosted SC island (with either topological Majorana or quasi-Majorana). We emphasize that the analysis above is valid if $\nu \gg \Gamma_R$; otherwise the transport mimics that of a normal MT. In this sense, a crossover between the normal and anomalous conductance features is anticipated via the tuning of $\nu$ or $\Gamma_R$. For instance, if $\max(T,eV) \ll \Gamma_R \ll \nu$ initially, we anticipate to experimentally observe the crossover from the high-order power law feature $G \sim [\max(T,eV)]^6$ to a constant conductance via increasing the value of $\Gamma_R$. We also emphasize that to observe these anomalous power laws and the crossover between them, the background zero-energy conductance ~$\Gamma_{L,R}/E_c$ or ~$\Gamma_{L,R}/\Delta_{sc}$ must be small, where $\Delta_{sc}$ refers to the superconducting gap.

When we tune the voltage to a different location $\delta_g = 1$, electron states $N$ and $N+2$ are degenerate and form the 2$e$ CB conductance peak[30,33] (also refer to SI (see Supplemental Material for more details.)). We notice that the 2$e$ peak height keeps almost constant in the relevant regime of this paper (i.e., $T \ll \Delta_{sc}, E_c$). The 2$e$ peak height is also very small compared to the largest 1$e$ peak (i.e., the 1$e$ peak value when $E \sim \nu$). Therefore, when plotting the 1e and 2e conductance peak values in the same figure, there expect to be two crossover points, as shown in Supplementary Fig.1, since 1e peak is non-monotonic (Fig. 1b) in our model. Indeed, the ratio between the maximum 1$e$ peak and the 2$e$ peak values equals $\Delta_{sc}/(gT)$[33] for the standard MT limit $\Delta_{sc} \gg T \gg \nu$, where $g \ll 1$ is the dimensionless tunneling conductance. Nevertheless, a 2$e$ conductance peak is expected to have little influence on a 1$e$ peak, since they are well-separated along the $n_g$ axis, with a large enough charging energy $E_c$.

## Methods
### Conductance evaluation

Here, we outline the derivations of the current operator and conductance. In our system, a bare tunneling at the left side, i.e., $-2\lambda_L \sum_k c_{kL}^\dagger S_z f_1$ is energetically forbidden, as it connects two states with different energies. However, with the Schrieffer-Wolff transformation[54], it can be used to construct high-order virtual tunnelings

$$\mathcal{O}_A = \sum_{p,q,k} \lambda_L c_{pL}^\dagger S_z f_1 \frac{1}{\nu - \epsilon_p} \lambda_R f_1^\dagger c_{kR} \frac{1}{\nu + \epsilon_q} \lambda_L c_{qL}^\dagger S_z f_1$$
$$\approx \sum_{\epsilon_p, \epsilon_q, k} \frac{2(\epsilon_p - \epsilon_q)}{\nu^3} \lambda_L^2 \lambda_R c_{pL}^\dagger c_{qL}^\dagger c_{kR} f_1, \tag{12}$$

to the leading order the energy difference $\epsilon_p - \epsilon_q$. This quantity is proportional to the temperature or bias at low energies.

To the leading order of the RG-irrelevant operator $\mathcal{O}_A$, the current operator approximately becomes

$$\hat{I} = ie\left[\mathcal{O}_A + \mathcal{O}_A^\dagger f_1^\dagger f_1 + \sum_k c_{kR}^\dagger c_{kR}\right] \equiv 2ie(\mathcal{O}_A - \mathcal{O}_A^\dagger). \tag{13}$$

We calculate the current at zero temperature, with a bias $V$ applied to the right lead. This bias can be incorporated as a time-dependent phase via the transformation $c_{kR}^\dagger \to c_{kR}^\dagger \exp(ieVt)$ (see e.g., refs. 60, 63).

With this trick, to the leading order of $t_L$, the current can be evaluated with Eq. (6). The full expression of the conductance then becomes

$$G = \frac{e^2}{\hbar} \frac{32\Gamma_L^2\Gamma_R^4}{3\pi^2\nu^6} \left[ -8\chi^2 + \chi(9+\chi^2)\arctan(\chi) + \right.$$
$$\left. + (-1+3\chi^2)\ln(1+\chi^2) + \tfrac{3}{2}\chi^2 \text{Li}_2(-\chi^2) \right], \quad (14)$$

where the energy ratio $\chi \equiv eV/\Gamma_R$, and $\text{Li}_n$ refers to the polylogarithm function. With Eq. (14) we plot Fig. 3. In two limiting cases, the conductance Eq. (14) approximately reduces to the results displayed in Eqs. (7) and (8).

## Data availability
Data sharing is not applicable to this article, as no datasets were generated or analyzed during the current study.

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

## Acknowledgements

The authors thank Zhan Cao, Xin Liu, and Hao Zhang for helpful discussions. We also thank Chung-Ting Ke for helpful discussion on the precision of experimental measurement. The work is supported by the Natural Science Foundation of China (Grants No. 11974198) and Tsinghua University Initiative Scientific Research Program.

## Author contributions

Y.H., G.Z., and D.L. did the calculation. G.Z., D.E.L., and Y.H. prepared the manuscript. D.E.L. initialized and supervised the project. All the authors contribute to the result analysis.

## Competing interests

The authors declare no competing interests.
