## [Peer Review File · Nature Communications]

Reviewers' comments:

Reviewer #1 (Remarks to the Author):

The Coulomb blockade is one of the experimental approaches to distill the non-locality of the Majorana-hosted superconducting island. The authors went beyond the Fu-teleportation with Majorana zero modes by studying the tunneling process between one Majorana zero mode and hybridized Majorana modes. They found the $1e$ tunneling conductance peak exhibits two distinct types of energy scaling in the low-energy limits. On the other hand, in the high-energy limit, the conductance peak is proportional to the inverse of the energy.

The manuscript shows interesting transport features of a topological superconductor island. If experimentalists can confirm their prediction, this theoretical study can potentially be a key proposal revealing the Majorana non-locality. The non-locality is an important stepping-stone to building Majorana-hosted quantum computers. However, the current proposal has a weak link connecting to the present experiments. Although the manuscript is well-written, I suggest the current version of this manuscript be transferred to a specialized journal. The reason in detail is in the following two paragraphs. If the authors can provide convincing reasons that the measurement of this conductance scaling is feasible in reality, I can reconsider suggesting the revised manuscript for Nature Communication publication.

The conductance peak of this so-called “double Fu-teleportation” stems from the high-order tunneling process. To be specific, in the low-energy limit, the conductance is too small, so experimentalists might not be able to confirm the power-law scaling predicted by the authors. The log plot of Fig.3 supports this concern (The conductance of the red line is much larger than the one of the blue line). Therefore, I strongly suggest the authors show the estimation of the conductance values by adopting the realistic values of the experimental parameters (say ref. [11]). Hopefully, the estimation can convince us that the current experimental setup can have this precision to measure the conductance for this proposed platform.

Fig. 1b is misleading. I strongly suspect any experiment can obtain the relation between the conductance peak and the energy like Fig. 1b. First, this panel is not a log plot since the green line is curved. Hence, the red line should not be straight. Second, in the low-energy limit, Eq. 7 and 8 show the conductance values are proportional to $(eV/\nu)^6$ and $(eV/\nu)^3$ respectively with the condition $(eV/\nu) \ll 1$. Therefore, the conductance peaks are extremely small. By comparing Eq. 11, the conductance in the high-energy limit is much greater than the one in the low-energy limit.

Hence, if Fig. 1b is not a log plot, the blue and red lines should be very close to zero conductance by comparing the green line.

There are some suggestions in the following to help the authors improve the manuscript.

1. The two coupled Majorana states together are effectively an Andreev bound state with small finite energy. Therefore, it is not surprising that the conductance peak exhibits a similar scaling when an Andreev bound state replaces the two hybridized Majorana modes. Perhaps using an Andreev bound state in the platform is much easier to intentionally set up the hybridization of the Majorana modes in experiments.

2. In the discussion section, the authors mentioned that the 2e conductance peak height is very small compared to the 1e peak. However, I am unsure if it is true in the low-energy limit. The conductance peaks in Eqs. 7 and 8 are also small. I am wondering if the authors can show the comparison of the conductance peak in 1e and 2e tunneling in different energy regimes.

3. What are ϵ_p and ϵ_q ? Their definitions are missing in the manuscript.

4. It is unclear that $\Gamma_L \gg \Gamma_L^{\text{eff}}$ with low-enough energies. Can the authors elaborate it?

Reviewer #2 (Remarks to the Author):

Dear Editor,

In this work, the authors "study the two-terminal transport through a CB island that hosts a MZM and two coupled MZMs (with the coupling amplitude ν) at opposite sides of the island". The authors show that this system of Majorana zero modes (MZMs) "displays unique transport features" in the

form of a cross-over between two power-law scaling from V^6 to V^3 as the applied voltage V crosses the normal state tunneling rate of one of the barriers. To show this, the authors use a transformation used in reference 32 to combine the Majorana operator and charge tunneling into a fermion operator f_1 . They then define another spin operator S which combines the Majoranas on the island in a way similar to the spin operator in Ref 50. What is interesting about the present work as opposed to transport works on the topological Kondo effect [48,50,51] is that these works are off-resonant and involve three terminals. In this regard, the authors should take note of the fact that PHYSICAL REVIEW B 99, 014512 (2019) studies resonant three terminal transport and is closer to the current work in this regard than [48,50,51]. Two terminal transport through an island with 4 Majoranas has also been previously discussed in PHYSICAL REVIEW B 92, 020511(R) (2015), where it was pointed out in a discussion that elastic co-tunneling can occur through a pair of localized Andreev states in a way that is quite similar to the Majorana case discussed in Ref 32. While the authors have missed to discuss some of these relevant references, I agree with the authors motivational statement that it is not "yet known whether or not the two-terminal CB island could provide a hallmark for non-local topological features of a Majorana-hosted SC island." My central concern is that I am not sure if the present manuscript resolves this issue. In the modern parlance of the field, the configuration studied in Fig 1c represents exactly the concern raised by Refs. [44-47]. So a system of this kind could very well show the transport signatures suggested in this work. It seems the authors call this system a "Majorana-hosted island" though reference [44-47] call this non-topological Andreev bound states to "quasi-Majorana".

The equivalent set-up in Fig 1d seems more sophisticated. However, at this point there doesn't seem to be a significant advantage to a two terminal set-up and one could just use the 2019 paper I mentioned earlier. In either of these cases, I am not sure how this transport signature can rule out additional Majorana modes in the lower segment of Fig 1d.

In addition to the above questions about the broader significance of this work I have concerns about the authors use of the word "Fu-teleportation" and "double Fu-teleportation". Despite Liang Fu's seminal contribution to the topic, the terminology "Fu-teleportation" disregards the literature that laid the steps for Ref 32, which is presumably the reason the authors use this terminology. The terminology of teleportation in the Majorana context was first introduced by Semenoff and Sodano cited as Ref 21 in Liang Fu's work (Ref 32). Additionally references 6,7 and 8 in Liang Fu's paper discussed transport signatures of this process. Liang Fu's critical contribution was to realize the importance of charging energy for the transport signature. However, teleportation as defined in terms of the Green function already existed in Semenoff and Sodano's work. Therefore, I do not agree with the terminology "Fu-teleportation". The terminology "double Fu-teleportation" is problematic because it suggests that this is as topological as "Fu-teleportation". However, as shown in the 2015 reference, configurations such as the one in Fig. 1c with non-topological localized end states can likely show an analog of "double Fu-teleportation". Therefore, the authors need to clarify what is unique or topological about this process that the authors make a central emphasis of this work.

In summary, while I think that the present work is a technically interesting

calculation, I do not see how it addresses the question raised in the introduction i.e. providing a hallmark of non-local topological features. As elaborated at the end of the first paragraph, this is because Fig 1c and other systems considered here are "non-topological" for most of the field. Also I expect superconducting islands with other less interesting couplings to show similar or similarly exotic power-laws in two terminal transport. Because of this I am not convinced of the significance of this work to a community beyond those who are interested in different power-laws in mesoscopic transport.

Response to the Referee Report

RESPONSE TO REFEREE A

We divide the report of Referee A into seven items (A1 to A7), and respond to them item by item.

A1. The manuscript shows interesting transport features of a topological superconductor island. If experimentalists can confirm their prediction, this theoretical study can potentially be a key proposal revealing the Majorana non-locality. The non-locality is an important stepping-stone to building Majorana-hosted quantum computers. However, the current proposal has a weak link connecting to the present experiments. Although the manuscript is well-written, I suggest the current version of this manuscript be transferred to a specialized journal. The reason in detail is in the following two paragraphs. If the authors can provide convincing reasons that the measurement of this conductance scaling is feasible in reality, I can reconsider suggesting the revised manuscript for Nature Communication publication.

We thank Referee A for considering our work as “interesting transport features”, “potentially be a key proposal”, and “well-written”. We are also grateful for all the questions from Referee A. They greatly help the improvement of our manuscript. We hope our responses below can satisfactorily answer the questions of Referee A.

A2. The conductance peak of this so-called “double Fu-teleportation” stems from the high-order tunneling process. To be specific, in the low-energy limit, the conductance is too small, so experimentalists might not be able to confirm the power-law scaling predicted by the authors. The log plot of Fig.3 supports this concern (The conductance of the red line is much larger than the one of the blue line). Therefore, I strongly suggest the authors show the estimation of the conductance values by adopting the realistic values of the experimental parameters (say ref. [11]). Hopefully, the estimation can convince us that the current experimental setup can have this precision to measure the conductance for this proposed platform.

We thank Referee A for this question that is significant for the experimental realization of our theoretical predictions. To provide a comprehensive picture, we answer this question by visiting three sub-questions, including: (i) What are the reasonable values of tuning parameters to observe our predicted features? (ii) With these chosen parameters, is the corresponding conductance experimentally accessible? (iii) What are the sub-leading tunneling events? And will they sabotage our theoretical predictions?

Before moving to detailed responses, we would like to first clarify the prerequisites of our theory. In our theory, we focus on the $1e$ peak, where the filling number n_g is fixed as a half-integer. Our system contains seven relevant energy scales: the temperature T , the bias eV , the charging energy E_c , the superconducting gap Δ_{sc} , the Andreev bound state energy ν , and two level broadening parameters Γ_L and Γ_R that quantify the Majorana-lead hybridization. In this work, we consider a realistic weak island-lead coupling regime with $(E_c, \Delta_{sc}) \gg \nu \gg (eV, T, \Gamma_L, \Gamma_R)$ in low-energy where (in the manuscript) we predict the crossover between different conductance power-laws. Here the

large E_c assumption guarantees the validity to focus on the situation with a half-filling Majorana island (i.e., with n_g a half-integer). The large gap Δ_{sc} assumption forbids transport via quasiparticle excitations above the superconducting gap. Following the data of Ref. [11] (or Ref. [10] of the latest manuscript), the charging energy E_c of a nanowire with radius 100nm and length 790nm is enough to produce a charging energy $E_c = 0.14\text{meV}$ (1.62 Kelvin in temperature). This charging energy can become even larger ($\sim 10\text{K}$), considering the technique advances that have enabled the fabrication of nanowires with a smaller radius $\sim 25\text{nm}$ [e.g., arXiv:2205.06736]. The induced superconducting gap is near the Al bulk gap, i.e., $\sim 0.2\text{meV}$ (equivalently $\sim 2.3\text{K}$). The values of both E_c and Δ_{sc} obtained from Ref. [11] are much larger than the experimentally accessible temperature $\sim 50\text{mK}$ reported in e.g., Ref. [63] of our manuscript. The hybridization strength ν between the two quasi-Majoranas γ_1, γ_2 on the left side of the nanowire can be tuned by changing the smoothness of the Gaussian-shape potential on the left. Indeed, the value of ν becomes larger for a sharper potential at the boundary. In addition, the hybridization strength ν is also a function of the chemical potential μ and the Zeeman energy V_z [1–3]. By adjusting μ and V_z properly, one can obtain a satisfactory value ν . Finally, $\nu \gg eV, \Gamma_L, \Gamma_R$ forbids the standard Majorana-assisted teleportation (we change the name from “Futeleration” to “Majorana-assisted teleportation” following the suggestion of Referee B): now the double Majorana-assisted teleportation becomes the leading process. We emphasize that parameters Γ_L, Γ_R, T and ν are all inessential parameters that are tunable experimentally. The precision of measurement (i.e., the experimental accessibility of the small conductance values) will be discussed in the following paragraphs.

With the above reminder of the prerequisites of our theory, we begin to answer the sub-question (i). To start with, in Fig. 3 of the previous manuscript, we choose a large range of bias in both limits to highlight the conductance power-laws of two limits, and the crossover between them. This wide-range choice of bias leads to a large difference in conductance in opposite limiting cases. This large range of bias is however unnecessary to detect the change of power-law features. Indeed, as shown in Fig. 3 of the latest manuscript, after the reduction of the bias range, the transition between different power-laws remains, while the difference of conductance in opposite limits has been significantly reduced. More specifically, by choosing parameters $\nu/\Gamma_L = 3, \nu/\Gamma_R = 25$, we can see the clear power-law scaling and the crossover behaviors in numerics. This numerical results of the conductance in the large bias limit ($eV \gg \Gamma_R$) is around $\sim 0.01e^2/h$, and that in the weak bias limit ($eV \ll \Gamma_R$) becomes $\sim 10^{-7}e^2/h$. We emphasize that the small value issue here is commonly encountered in all high order power-law behaviors, where the leading irrelevant processes have a higher scaling dimension.

Now we move to discuss sub-question (ii) on the detail values regarding the experimental accessibility of these conductance. Firstly, the large-bias conductance $\sim 0.01e^2/h$ (with $G \sim \text{Max}[T, eV]^3$) is clearly observable. Indeed, it has been shown in e.g., Refs. [63,65] that even a smaller conductance $\sim 10^{-3}e^2/h$ can be measured with enough precision (enough to display the power-law) in both a nanowire system and a two dimensional electron gas system. In the opposite limit, the conductance $\sim 10^{-7}e^2/h$ is indeed rather small. Fortunately, conductance of this precision has been realized in Ref. [66] when measuring the conductance quanta. As another example, in the supple-

mentary of Ref. [68], it is shown that a small transparency at this order ($\sim 10^{-7}$) can be obtained with a spectroscopy measurement, while keeping a clear power-law feature. In Ref. [67], the power-law feature $G \propto T^\alpha$ has also been observed in long InSb nanowires (with the power $2.3 < \alpha < 4.6$). In these systems, the observed smallest conductance (with visible power-law features) can be as small as around $10^{-11}e^2/h$, which is perfectly enough for the observation of the power-law feature predicted by us. Therefore, in low-energy regimes, we anticipate conductance as experimentally accessible in both limits (both $\sim 10^{-2}e^2/h$ and $\sim 10^{-7}e^2/h$). We emphasize that (following evidence above) if we only want to observe the crossover and the $G \sim \text{Max}[T, eV]^3$ scaling regimes, or exotic non-monotonic temperature-dependence (at even higher energy with much larger conductance values) of the Coulomb blockade peak, the requirements are significantly relaxed. The $G \sim \text{Max}[T, eV]^6$ scaling regime is nontrivial but allowed by current start-of-the-art technology.

Finally, we move to the last issue (iii), and show the amplitude of interruptions from other sources. There can in-general exist two categories of tunneling processes that might potentially sabotage the predicted non-local features. Category of the first kind involves processes that can be suppressed after a careful choice of experimental parameters. For instance, particles might sequentially tunnel through local states induced by e.g., impurities and irregularities of the sample. These processes, fortunately, are high-order processes that decay exponentially as a function of the size of the Majorana island. They can be thus removed via increasing the island size (e.g. nanowire length). Similarly, particles might also tunnel via quasiparticle states above the superconducting gap. These tunneling processes are instead strongly disfavored by choosing a large enough superconducting gap. As the second category, interruption might arise from sub-leading tunneling processes. Since local tunnelings (which consists of sequential local tunnelings through the entire system) are suppressed (by the Majorana-island size), in our work the sub-leading processes can be the descendents of the leading operators. These operators, however, are either produced after the RG flow, or constructed as higher-order processes. Their initial amplitudes are thus negligible. In addition, these descendent operators have larger scaling dimensions in the renormalization group perspective, in comparison to the leading operators. The descendent operators thus become less and less important after the renormalization group flow, and remain negligible at low energies. One might also anticipate possible interruption from $2e$ conductance. This interruption is fortunately negligible, as $1e$ and $2e$ conductance dominate under quite different choices of the backgate voltage (see the response to A5 below).

Following analysis above, the conductance required to prove our theory is within the range of experiment, and will not be sabotaged by interruptions from higher-order processes. We thus anticipate our theoretical predictions as experimentally accessible.

A3. Fig. 1b is misleading. I strongly suspect any experiment can obtain the relation between the conductance peak and the energy like Fig. 1b. First, this panel is not a log plot since the green line is curved. Hence, the red line should not be straight. Second, in the low-energy limit, Eq. 7 and 8 show the conductance values are proportional to $(eV/\nu)^6$ and $(eV/\nu)^3$ respectively with the condition $(eV/\nu) \ll 1$. Therefore, the conductance peaks are extremely

small. By comparing Eq. 11, the conductance in the high-energy limit is much greater than the one in the low-energy limit. Hence, if Fig. 1b is not a log plot, the blue and red lines should be very close to zero conductance by comparing the green line.

We appreciate Referee A for pointing out the confusing plot of Fig. 1b. Indeed, considering its smallness, the value of conductance along the blue curve is only visible if displayed with a log-log plot. In the latest manuscript we have thus replotted Fig. 1b in a log-log way to avoid possible confusion.

A4. The two coupled Majorana states together are effectively an Andreev bound state with small finite energy. Therefore, it is not surprising that the conductance peak exhibits a similar scaling when an Andreev bound state replaces the two hybridized Majorana modes. Perhaps using an Andreev bound state in the platform is much easier to intentionally set up the hybridization of the Majorana modes in experiments.

We thank Referee A for this important suggestion. Indeed, our theory applies to situations where the left side contains either two coupled Majoranas, a pair of coupled quasi-Majoranas, as well as a finite-energy Andreev bound state. We only require a large enough energy of the impurity state that couples to the left lead. As for the right side, the right lead should couple to an isolated Majorana (either a topological Majorana or one decoupled quasi-Majorana) to enable non-local teleportation, i.e., the Majorana-assisted teleportation.

A5. In the discussion section, the authors mentioned that the $2e$ conductance peak height is very small compared to the $1e$ peak. However, I am unsure if it is true in the low-energy limit. The conductance peaks in Eqs. 7 and 8 are also small. I am wondering if the authors can show the comparison of the conductance peak in $1e$ and $2e$ tunneling in different energy regimes.

We thank Referee A for this important question. In the latest manuscript, we have added another section (Sec. VI) together with a plot (Fig. S1) to discuss this issue in the supplementary materials. As can be seen from Fig. S1, the $1e$ peak conductance (the black curve) is not always larger than the $2e$ one (the blue curve). Indeed, a more subtle investigation shows that the $1e$ conductance peak becomes larger than that of $2e$, when $\Gamma_R \lesssim E \lesssim \Delta_{sc}^2/\Gamma$. In the previous manuscript, we are only stating that the largest $1e$ peak conductance (the value around $E = \nu$) is larger than $2e$ conductance peak value (see Fig. S1 of the latest manuscript). In the latest manuscript, we have added several lines in the section ‘‘Discussion’’ to clarify this point.

Although $2e$ conductance peak can exceed $1e$ peak in certain energy regimes, we emphasize that $1e$ conductance peak, which is our major topic, will not be interrupted by $2e$ conductance signals, since $2e$ and $1e$ peaks emerge at completely different backgate voltages. Indeed, $1e$ and $2e$ conductance peaks requires half filling ($n_g = 1/2$) and integer filling ($n_g = 1$), respectively. With a large charging energy (which is our assumption), $1e$ and $2e$ conductance peaks are well separated along the n_g axis. As the consequence, $1e$ and $2e$ peaks, no matter which one is larger, will not influence the observation of each other.

A6. What are ϵ_p and ϵ_q ? Their definitions are missing in the manuscript.

We thank Referee A for pointing out this obscurity. Basically, ϵ_p and ϵ_q refer to the energy of particles with

momenta p and q , respectively. We have added their clearer definition after Eq. (4) of the latest manuscript.

A7. It is unclear that $\Gamma_R \gg \Gamma_L^{\text{eff}}$ with low-enough energies. Can the authors elaborate it?

We thank Referee A for pointing out this obscurity. In our work, Γ_L^{eff} refers to the level broadening induced by the high-order transmission \mathcal{O}_A of Eq. (4). As a higher-order tunneling operator, the prefactor of \mathcal{O}_A is comparatively smaller than the direct hybridization at the right side, i.e., Γ_R . In addition, since \mathcal{O}_A contains three fermionic operators (for fermions in the lead), it has a larger scaling dimension ($\alpha = 3/2$) in comparison to that of the hybridization at the right-hand-side (Γ_R has $\alpha = 1/2$). From the renormalization group point of view (see e.g., Ref. [57]), the prefactor of \mathcal{O}_A , i.e., Γ_L^{eff} becomes less and less important at lower energies. By contrast, the hybridization between the right lead and the Majorana next to it becomes more and more important, effectively leading to $\Gamma_R \gg \Gamma_L^{\text{eff}}$ at low energies.

In the latest manuscript, we have added several lines in the second paragraph after Eq. (4) to elaborate our point above.

I. RESPONSE TO REFEREE B

We divide the report of Referee B into eight items ($B1$ to $B8$), and respond to them item by item.

B1. In this work, the authors "study the two-terminal transport through a CB island that hosts a MZM and two coupled MZMs (with the coupling amplitude ν) at opposite sides of the island". The authors show that this system of Majorana zero modes (MZMs) "displays unique transport features" in the form of a cross-over between two power-law scaling from V^6 to V^3 as the applied voltage V crosses the normal state tunneling rate of one of the barriers. To show this, the authors use a transformation used in reference 32 to combine the Majorana operator and charge tunneling into a fermion operator f_1 . They then define another spin operator S which combines the Majoranas on the island in a way similar to the spin operator in Ref. [50].

We thank Referee B for summarizing our work. We are also grateful about all questions from Referee B. These questions help a lot on the improvement of our manuscript.

B2. What is interesting about the present work as opposed to transport works on the topological Kondo effect [48,50,51] is that these works are off-resonant and involve three terminals. In this regard, the authors should take note of the fact that Physical Review B 99, 014512 (2019) studies resonant three terminal transport and is closer to the current work in this regard than [48,50,51]. Two terminal transport through an island with 4 Majoranas has also been previously discussed in Physical Review B 92, 020511(R) (2015), where it was pointed out in a discussion that elastic co-tunneling can occur through a pair of localized Andreev states in a way that is quite similar to the Majorana case discussed in Ref. [32].

We thank Referee B for pointing out these references. We agree that these references are indeed highly relevant to our current work. Citations (2019 paper as Ref. [54] of the latest manuscript) to them have been added in the latest manuscript. Nevertheless, our work tackles with problems that are quite different from these references. Yes, our

setup involves only two terminals as pointed out by referee B.

As for the 2015 paper (Ref. [38] of the latest manuscript), its Andreev (topologically trivial regime) case consists of two Andreev bound states with zero energies. By contrast, we consider one Andreev bound state (the one next to the left lead) with a finite energy ν along with a single Majorana mode on the other side. In our theory, this finite energy ν is the very ingredient that generates the crossover between two different polynomial conductance features, which manifest the non-local transport. See more details in the response to *B3* and *B8* below.

B3. While the authors have missed to discuss some of these relevant references, I agree with the authors motivational statement that it is not "yet known whether or not the two-terminal CB island could provide a hallmark for non-local topological features of a Majorana-hosted SC island." My central concern is that I am not sure if the present manuscript resolves this issue.

We thank Referee B for this very important question. Before answering this question, we would like to emphasize that our calculation aims mainly on the disclosure of the non-locality of Majorana-assisted teleportation (we change "Fu-teleportation" to "Majorana-assisted teleportation" following your suggestion). However, our proposal does not aim to distinguish quasi-Majorana from topological ones, as Majorana of either category can generate non-local transport across the Majorana island. It is also claimed by e.g., Ref. [47] that quasi-Majoranas also suffices to realize Majorana braiding.

As a reminder, in our work the non-locality is predicted to be manifested by two crucial features, i.e., (i) the fact that conductance depends polynomially on the bias (power-laws), and (ii) the crossover between two polynomial features as energy decreases. Below we compare both features to that in systems with only local transport, through which we show that these two features together provide a strong evidence of non-locality.

To start with, in our system, the energies are well below the superconducting gap, thus forbidding transport (through the island) via real quasiparticle states above the gap. Consequently, local transport can only rely on either virtual quasiparticle states, or sequential tunnelings via localized impurity states. These two local transport options (through virtual states), however, are suppressed greatly by either the superconducting gap, the charging energy, or the size of the Majorana island, leading to an exponentially suppressed (instead of polynomially featured) conductance at low energies. As the consequence, this polynomial conductance feature (versus the exponential feature of local transport) at low energies is a strong signature of non-locality. We are aware of the fact that polynomial conductance features also appear in some other local-transport-only systems, e.g., the two-channel Kondo model (e.g., Ref. [62]) or dynamical Coulomb blockade (e.g., Ref. [63]) systems. However, we would like to emphasize that in these systems (that have only local transport and polynomial conductance features), the involved bulk systems (leads) are all metallic (gapless), instead of a fully gapped superconducting bulk of our case. As another example, the coexistence of polynomial conductance features and non-locality also exists in e.g., PRB 57, 9879 (1998) by D. L. Maslov and P. M. Goldbart, where non-local transport is instead enabled by non-local Andreev reflection. Such non-local feature, in contrast to our case, will however disappear when the wire length increases (to be larger than the superconducting

FIG. 1: Comparison of transport with non-local Majorana state of our system and the case with only local states. Also see Fig.1(d) of the revised main text.

coherence length).

Now we move to the second point, i.e., the crossover between two limiting cases that display distinct polynomial features (indicated by two dashed lines of Fig. 3). Most importantly and counter-intuitively, in our theory the universality class of the transport (manifested by the conductance power-law), which is a global feature, alters after the hybridization between the right lead and the nonlocal state f_1 (from Majorana modes γ_1 and γ_4). In fact, the crossover can be understood from a simple scaling analysis of our Equation (4) in the main text, that is the the process A operator

$$\mathcal{O}_A = \sum_{\epsilon_p > \epsilon_q, k} \frac{2(\epsilon_p - \epsilon_q)}{\nu^3} \lambda_L^2 \lambda_R c_{pL}^\dagger S_z f_1 \cdot f_1^\dagger c_{kR} \cdot c_{qL}^\dagger S_z f_1. \quad (1)$$

At very low energies, the coupling between the state f_1 (nonlocal fermion) and the right lead is marginal in the RG sense, therefore, the f_1 operator in A operator is fully absorbed by the right lead, and acquires the scaling dimension as an effectively free fermion in the right lead. So, we get a larger power-law scaling for the low-energy conductance (power counting of \mathcal{O}_A operator). On the other hand, at higher energies (T or eV), the tunneling between f_1 state and the right lead loses quantum coherence, and then the f_1 operator becomes local impurity state and loses scaling dimension. In that case, the conductance shows a smaller power-law scaling (power counting of \mathcal{O}_A operator). This **right lead** – non-local state coupling behavior mentioned above is one of the key ingredients for the change of the power-law feature. The other key ingredient comes from the Majorana coupling ν **on the left hand side**. You can see that the **right lead** hybridization needs to combine with the **left** Majorana coupling to generate this novel feature. Indeed, these two causing ingredients are essentially “non-local” (one from right and the other from left). Therefore, we believe this non-local nature of the Majorana system can be manifested by our predicted unique features, i.e., power-laws and the crossover between them. The intuitive picture is shown in Fig. 1 above [also Fig. 1(d) of the latest main text].

In addition, this feature is highly non-trivial since normally (including the example mentioned above that involves

nonlocal Andreev reflections) the hybridization of a local state would not greatly change the global transport feature, especially when the transport is through a gapped superconducting bulk. This highly non-trivial event however occurs in our case, since the Majorana state γ_4 , either topological or quasi, plays a special and irreplaceable role in non-local transport. The change of its status (through the hybridization of f_1 that consists of γ_4) thus naturally induces a strong (qualitative) modification of the transport feature (the upper panel of Fig. 1). By contrast, for cases where transport relies on sequential local tunnelings (the lower panel of Fig. 1), the hybridization between the right lead and the nearby local site will introduce only a minor change. Indeed, as has been stated in e.g., Ref. [58] for the Kondo model, the hybridization between the lead and a single impurity state just effectively extends the lead, at the expense of an extra phase shift. We once again emphasize that our system is different from other local-transport systems that can also host a transition of universality class (e.g., the two channel Kondo model Ref. [62]). In these cases the impurity(s) bridges two wires or reservoirs that are conducting instead.

Finally, Referee B has the concern that some other systems might display the same or similar features predicted by us, thus potentially undermining our evidence of nonlocality. The concern involves Refs. [44-47] (of the previous manuscript) for quasi-Majoranas and Refs. [48,50,51] (of the previous manuscript) for topological Kondo systems. As the starter, we agree that indeed Refs. [44-47] with quasi-Majoranas can display the predicted transport features, as quasi-Majoranas can also support nonlocal transport. We thank Referee B for pointing it out. As we have stated at the beginning of the response to B3, our work does not aim to distinguish quasi-Majorana from topological ones, but to highlight the non-locality of Majorana-assisted teleportation. In the latest manuscript, to avoid possible misunderstanding, we have added several lines to more clearly state this fact (see the parentheses at the end of the second paragraph and in the section “Discussion”, and also the lines above the section “Results”). This fact, however, does not undermine the significance of our work. Indeed, quasi-Majoranas are predicted (Ref. [47]) to be also applicable to braiding. Distinguishing quasi-Majoranas from trivial ABSs to-date remains a non-trivial and meaningful task. Now we move to the topological Kondo cases of Refs. [48,50,51]. As the first remark, topological Kondo effect also displays polynomial conductance features, as it also supports non-local transport. In this sense, the polynomial conductance feature of a topological Kondo system does not contradicts one of our summaries that polynomial conductance feature through the superconducting island can signify the non-local feature. There are however two important differences between topological Kondo effect and our proposal. As the first difference, a topological Kondo system requires three or more leads. Each lead couples to one Majorana to detect the non-locality. This is experimentally more complicated than our proposal where only two leads are required. As the second difference, as far as we know, change of polynomial conductance features of a topological Kondo model occurs only when the system flows between fixed points with different conductance values (see e.g., Fig. 3 of Ref. [53]). By contrast, in our system the modification of the polynomial conductance feature occurs when the system approaches the same fixed point (that has zero conductance). As we have analyzed above, this feature (the crossover between two power-law features) is itself another evidence of non-locality. More importantly, the crossover between two power-law features, as far as

we know, can only occur in Majorana-assisted teleportations, thus telling Majorana-assisted teleportation from other non-local transport processes. Indeed, the non-local Andreev reflection, which also occurs via non-local transport, do not have a crossover between two qualitatively different conductance features, by simply hybridizing a single site. Of this (non-local Andreev reflection) case, the hybridization of a single site simply reduces the effective width of the superconducting island.

Following discussions above, the non-local features manifest themselves in both the polynomial conductance features in two limiting cases, and the crossover between these. These two features together provide a strong evidence of the non-locality of the Majorana-assisted transport.

In the latest manuscript we have added parentheses before the section “Brief summary” and in the section “Discussion”, together with several lines before the section “Results” to clarify the purpose of the manuscript, i.e., to figure out signatures of non-locality. We have also added several lines before Eq. (1), and a paragraph before Eq. (6) to briefly discuss why conductance features predicted by our manuscript convey information of non-locality. A figure (a new Fig. 1d) is also added to better illustrate our point.

B4. In the modern parlance of the field, the configuration studied in Fig 1c represents exactly the concern raised by Refs. [44-47]. So a system of this kind could very well show the transport signatures suggested in this work. It seems the authors call this system a "Majorana-hosted island" though reference [44-47] call this non-topological Andreev bound states to "quasi-Majorana".

We agree with Referee B that our predictions are valid for a system that hosts either a topological Majorana, or a pair of decoupled quasi-Majoranas, i.e., the situations discussed in Refs. [44-47] (of the previous manuscript). Indeed, as has been mentioned in the response to B3, our manuscript focuses on the manifestation of non-locality of transport. The non-local feature, however, occurs for transport through either a topological Majorana, or quasi-Majoranas.

B5. The equivalent set-up in Fig. 1d seems more sophisticated. However, at this point there doesn't seem to be a significant advantage to a two terminal set-up and one could just use the 2019 paper I mentioned earlier. In either of these cases, I am not sure how this transport signature can rule out additional Majorana modes in the lower segment of Fig. 1d.

Indeed, the same as the structure of Figs. 1a and 1c, the structure of Fig. 1d can not distinguish a quasi-Majorana from a topological one. In the latest manuscript, we have replaced Fig. 1d by another figure to indicate why the crossover between two polynomial features can manifest the non-local feature of transport.

B6. In addition to the above questions about the broader significance of this work I have concerns about the authors use of the word "Fu-teleportation" and "double Fu-teleportation". Despite Liang Fu's seminal contribution to the topic, the terminology "Fu-teleportation" disregards the literature that laid the steps for Ref. 32, which is presumably the reason the authors use this terminology. The terminology of teleportation in the Majorana context was first introduced by Semenoff and Sodano cited as Ref. 21 in Liang Fu's work (Ref. 32). Additionally references 6,7 and 8 in Liang Fu's paper discussed transport signatures of this process. Liang Fu's critical contribution was to

realize the importance of charging energy for the transport signature. However, teleportation as defined in terms of the Green function already existed in Semenoff and Sodano's work. Therefore, I do not agree with the terminology "Fu-teleportation". The terminology "double Fu-teleportation" is problematic because it suggests that this is as topological as "Fu-teleportation".

We thank Referee B to bring up this issue. After checking more references, we notice that the terminology "Fu-teleportation" is indeed not so frequently used. We have thus changed "Fu-teleportation" to "Majorana-assisted teleportation" in the latest manuscript. This terminology has been directly and indirectly used in e.g., Refs. [54] and [38].

We also thank Referee B for bringing the paper by Semenoff and Sodano to us. We agree that this is a pioneering paper that has established the foundation of Majorana-assisted teleportation. Reference to this paper has been added in the latest manuscript.

B7. However, as shown in the 2015 reference, configurations such as the one in Fig. 1c with non-topological localized end states can likely show an analog of "double Fu-teleportation". Therefore, the authors need to clarify what is unique or topological about this process that the authors make a central emphasis of this work.

As the start, we agree with Referee B and the 2015 reference authored by J. Sau et al that the non-local teleportation predicted by Liang Fu in 2010 is not a confirmative evidence of Majorana. Indeed, non-local teleportation can also occur through two quasi-Majorana pairs that reside at opposite sides of the wire.

However, we do not think this fact undermines our work, for reasons below. Firstly, the 4π -periodicity predicted by the 2015 paper is also known from topologically trivial effects, e.g., the Landau-Zener effect [PRL, 111, 046401 (2013) by M. Houzet et al]. Several prerequisites have also been proposed to observe this periodicity in experiment. As another fact, the double teleportation process predicted by the 2015 reference is not really the "double Fu-teleportation" predicted by our work. Indeed, in the 2015 reference, the authors comment that when two zero-energy Andreev bound states are located at opposite sides of a nanowire, their states can change simultaneously by absorbing two electrons after breaking a Cooper pair in the superconducting bulk. This is however not our case, as in our manuscript, the coherent double Majorana-assisted teleportation consists of the transfer of two charges between two leads, through which a persistent current can be produced. As another difference from the 2015 reference, the left Andreev bound state of our case has a finite energy ν . Indeed, this energy is assumed to be larger than both the bias and the level broadening in the more interesting low-energy regime. We emphasize that the finite value of ν is the crucial element that enables the coherent double Majorana-assisted teleportation. Otherwise, transport relies on single Majorana-assisted teleportation realized by two Majoranas (or quasi-Majoranas) that couple to each lead. Finally, in previous works (including the 2015 reference), no one has ever analyzed what happens after a strong lead-Majorana hybridization. In our manuscript, we have shown that the hybridization of the Majorana leads to a strong modification of the universality class. Importantly, this strong modification only occurs for a non-local transport (as in the response to *B3*), and thus becomes a strong signature of non-locality as we discussed in *B3*.

B8. In summary, while I think that the present work is a technically interesting calculation, I do not see how it addresses the question raised in the introduction i.e. providing a hallmark of non-local topological features. As elaborated at the end of the first paragraph, this is because Fig. 1c and other systems considered here are "non-topological" for most of the field. Also I expect superconducting islands with other less interesting couplings to show similar or similarly exotic power-laws in two terminal transport. Because of this I am not convinced of the significance of this work to a community beyond those who are interested in different power-laws in mesoscopic transport.

We thank Referee B for considering our work as interesting.

As in the response to B3, in this work we are trying to propose a setup that can generate experimentally accessible data to manifest non-local transport features. We do not, however, plan to distinguish quasi-Majoranas from topological ones, as both can mediate non-local Majorana-assisted teleportation, as has been mentioned by Referee B. Several lines have been added in the latest manuscript to emphasize and better clarify our purpose.

Nevertheless, we would like to emphasize again why power-law features and the crossover between two different power-law features are important in mesoscopic physics based on our knowledge. Firstly, and maybe most importantly to our current work, power-law features are normally absent in systems with a gapped bulk. Indeed, power-law conductance features most commonly occur in strongly correlated systems, e.g., a Kondo model, where a single or multiple impurities connect two or more reservoirs that are metallic. In systems with a gapped bulk, in strong contrast, a conductance with power-law feature is strongly associated to non-local, or "teleportation-like" transport, e.g., non-local Andreev reflection: otherwise transport through a gapped area is exponentially suppressed by the insufficiency of energy. In this sense, a power-law conductance feature through a gapped system is itself a strong non-local signature.

As the second importance (of power-law features), at low energies, the polynomial conductance feature tells us the information (the universality class) of the system ground state. For instance, in the two-channel Kondo model, a free Majorana mode can also be generated from quantum frustration at the quantum critical point. This impurity Majorana, interestingly, can be manifested by the power-law conductance feature, as has been stated by PRL 116, 157202 (2016) by A. K. Mitchell et al. Similarly, Ref. [63] uses the featured power-law as the experimental signature of the non-trivial degeneracy (i.e., the impurity entropy) residued at zero temperature, at the quantum critical point.

As an extension of the second point, the variation of the polynomial feature, which is among our major points, indicates a qualitative change of the system status. Indeed, following discussions around Fig. 2 of our manuscript, the leading transport operator greatly changes after the hybridization of γ_4 by the right lead. Notably, this qualitative modification occurs since γ_4 plays an important and irreplaceable role in the Majorana-assisted teleportation. The hybridization of γ_4 then greatly modifies the nature of transport process. As a strong contrast, for more conventional non-local transport like non-local Andreev reflection, the hybridization of a single site leads to only quantitative modification (of transport). Indeed, now the superconducting island simply becomes effectively narrower. The nature of non-local Andreev reflection remains invariant (after the hybridization of a single site).

As a summary, our polynomial conductance feature discloses the non-local feature. The crossover between two

different power-laws, on the other hand, shows that the state/site that couples to the right lead, i.e., γ_4 , is of special importance and plays a distinct role from other sites of the superconducting bulk. This special role of γ_4 indicates that γ_4 is a decoupled (either topological or quasi) Majorana.

LIST OF CHANGES MADE

Based on the questions and suggestions from the Referees, we have made several changes. We use the LaTeX diff tool to generate a file “**diff.pdf**” for comparing the difference between the original version and the revised version of the main text. In addition, we also summarize the major changes below.

- 1) We have changed all “Fu-teleportation” by “Majorana-assisted teleportation” in the latest manuscript.
- 2) We have added two parentheses (both at the end of the second paragraph, and in the “Discussion” section), to emphasize that our theory applies to both topological and quasi Majorana states. This point is emphasized again before the section “Results”: there comparison between the case of Ref. [38] is also provided.
- 3) In the paragraph before the paragraph that contains Eq. (1), we have added several lines to explain why a qualitative change of the transport feature can signify the non-local feature of the transport.
- 4) After Eq. (3) we have added several lines to compare our setup with the topological Kondo setups.
- 5) We have replaced the sample setup of Fig. 1(d) by a demo that more intuitively explains why the change of polynomial features can disclose the non-locality.
- 6) After Eq. (4) we have added two lines to give a clearer definition on ϵ_p and ϵ_q .
- 7) In the second paragraph after Eq. (4), we have added several lines to explain why we take $\Gamma_R \gg \Gamma_L^{\text{eff}}$ in our analysis.
- 8) In the second paragraph before Eq. (6), we have added a paragraph to explain why these two predicted conductance features can disclose the non-locality of transport.
- 9) After Eq. (8), we have added several lines on the experimental accessibility of our theory.
- 10) At the end of the section “Discussion”, we have added several lines to emphasize that $1e$ conductance peak will not be influenced by the $2e$ ones.
- 11) In the supplementary materials, we have added another section (Sec.VI: $2e$ conductance peak) together with a plot (Fig.S1) to discuss $2e$ conductance peak.

-
- [1] Christopher Moore, Chuanchang Zeng, Tudor D Stanescu, and Sumanta Tewari, “Quantized zero-bias conductance plateau in semiconductor-superconductor heterostructures without topological majorana zero modes,” *Phys. Rev. B* **98**, 155314 (2018).
 - [2] Christopher Moore, Tudor D Stanescu, and Sumanta Tewari, “Two-terminal charge tunneling: Disentangling majorana zero modes from partially separated andreev bound states in semiconductor-superconductor heterostructures,” *Phys. Rev. B* **97**, 165302 (2018).
 - [3] A. Vuik, B. Nijholt, A. R. Akhmerov, and M. Wimmer, “Reproducing topological properties with quasi-majorana states,” *SciPost Phys.* **7**, 061 (2019).

REVIEWERS' COMMENTS

Reviewer #1 (Remarks to the Author):

After reading the correspondences, I found that the authors answered all of my questions and clarified my concerns about the experimental feasibility. My major concern in my previous report is that the tunneling conductance in some energy regions might be too small to be experimentally measured. First, the authors estimated the different values of conductance in the different energy regions. It is not surprising that their estimated value of conductance is very small. However, the updated manuscript provides the current experimental progress on the conductance measurement and shows the precision measurement is much better than the requirement of this experimental setup. Furthermore, there is an additional concern about the sub-leading tunneling terms, which can destroy their proposed conductance scaling. Their reply also clearly ruled out this unwanted scenario. Hence, the conductance scaling in this proposed setup can be measured by the current state-of-the-art technology, and the observation of this scaling conductance can likely be a smoking gun of the Majorana non-locality.

Since the Majorana-assisted teleportation in the literature (e.g., Liang Fu's original proposal) is unable to provide undebatable evidence to support the non-locality of a Majorana-hosted SC island, this manuscript further improves the transport design and shows the Majorana non-locality can be revealed by the power law features of the $1e$ tunneling conductance. Since the scaling of the power law depends on the different energy regions, this sophistication might be able to rule out scenarios other than Majorana non-locality. In this regard, I believe this theoretical study can be an important proposed experimental design to reveal the Majorana non-locality, which is one of the pillars of building Majorana-hosted quantum computers. Furthermore, the manuscript clearly presents their main idea of the Majorana island design. Thus, I would like to recommend this manuscript for nature communication publication.

Reviewer #2 (Remarks to the Author):

Dear Editor,

In the revised version, in my opinion, the authors have addressed most of the concerns raised

by both referees. Specifically, with regard to my concerns, the authors have clarified that the scaling that they propose does not specifically distinguish Majorana and quasi-Majorana states. However, I agree with the authors that the manuscript proposes interesting power-law non-local transport in Majorana hosting islands that is distinct from Coulomb blockade.

That being said, I continue to have reservations about the terminology used that have been more solidified by the referees responses. The first concern is likely easy to address. The authors state that their signatures are not affected by the distinction between Majorana and quasi-Majorana. The authors then argue that since quasi-Majoranas have been proposed for use in braiding this distinction is not significant. My issue is that quasi-Majoranas are proposed to arise without topological superconductivity. So the phrase topological superconductivity in the title (and anywhere else in the manuscript) must be changed to Majorana-hosting island or something of the sort.

The second concern is with the use of the word teleportation. The authors state that "Indeed, non-local teleportation can also occur through two quasi-Majorana pairs that reside at opposite sides of the wire." and "we do not think this fact undermines our work...is also known from topologically trivial effects...Houzet et al.]." The Landau-Zener mechanism in Houzet et al does not apply to teleportation in Ref 39. In fact, the claim of Ref 39 is that non-local transport through quasi-Majorana differs from teleportation proposed in Ref 32 using the interferometric signal despite the simple transport properties being the same. This is because interferometry tests the coherence of the teleported electron. As the authors point out in the response, the current proposal is about a double teleportation process at low voltages. Since pairs of electrons are transferred, the interferometric signature will be 2π periodic similar to conventional superconductors.

The voltage dependence will of course be different. I understand that from Fig 1d, the analysis of the process uses a non-local Fermion. However, the coherent transport of single electrons through this level is not what is occurring in any case. So I propose the author qualify the term teleportation with some word e.g. double teleportation or Cooper pair teleportation. This raises some complications with what to do with Fig 2(ii) - but I leave that to the authors to decide. The main issue to be resolved is that the term teleportation should be modified to clarify that this process is not quite the same as the one discussed in Ref 32, which can distinguish Majorana from quasi-Majorana.

In summary I am happy to recommend the manuscript for publication as long as the authors can modify the manuscript to clarify the concerns.

Response to Referee A

We gratefully thank referee A for carefully reading our work, and their recommendations to publish our manuscript on Nature Communications.

Response to Referee B

In the revised version, in my opinion, the authors have addressed most of the concerns raised by both referees.

Our reply: We gratefully thank Referee B for the appreciation on our revised manuscript.

Specifically, with regard to my concerns, the authors have clarified that the scaling that they propose does not specifically distinguish Majorana and quasi-Majorana states. However, I agree with the authors that the manuscript proposes interesting power-law non-local transport in Majorana hosting islands that is distinct from Coulomb blockade. That being said, I continue to have reservations about the terminology used that have been more solidified by the referees responses.

The first concern is likely easy to address. The authors state that their signatures are not affected by the distinction between Majorana and quasi-Majorana. The authors then argue that since quasi-Majoranas have been proposed for use in braiding this distinction is not significant. My issue is that quasi-Majoranas are proposed to arise without topological superconductivity. So the phrase topological superconductivity in the title (and anywhere else in the manuscript) must be changed to Majorana-hosting island or something of the sort.

Our reply: We thank Referee B for this important suggestion. Indeed, our discussion applies to systems with either topological Majorana, or quasi-Majorana zero modes. In the latest manuscript, we have replaced “topological superconducting island” in the title by “Majorana-hosted superconducting island”. We have also carried out corresponding modifications on the caption of Fig. 2.

The second concern is with the use of the word teleportation. The authors state that "Indeed, non-local teleportation can also occur through two quasi-Majorana pairs that reside at opposite sides of the wire." and "we do not think this fact undermines our work...is also known from topologically trivial effects...Houzet et al.]." The Landau-Zener mechanism in Houzet et al does not apply to teleportation in Ref 39. In fact, the claim of Ref 39 is that non-local transport through quasi-Majorana differs from teleportation proposed in Ref 32 using the interferometric signal despite the simple transport properties being the same. This is because interferometry tests the coherence of the teleported electron.

Our reply: We thank Referee B for pointing out the way to distinguish between non-

local transport through quasi-Majorana and the teleportation through topological Majorana.

As the authors point out in the response, the current proposal is about a double teleportation process at low voltages. Since pairs of electrons are transferred, the interferometric signature will be 2π periodic similar to conventional superconductors. The voltage dependence will of course be different. I understand that from Fig 1d, the analysis of the process uses a non-local Fermion. However, the coherent transport of single electrons through this level is not what is occurring in any case. So I propose the author qualify the term teleportation with some word e.g. double teleportation or Cooper pair teleportation. This raises some complications with what to do with Fig 2(ii) - but I leave that to the authors to decide. The main issue to be resolved is that the term teleportation should be modified to clarify that this process is not quite the same as the one discussed in Ref 32, which can distinguish Majorana from quasi-Majorana.

Our reply: We thank Referee B for this important suggestion. In the latest manuscript, we call the non-local transport predicted by Ref. 32 as “topological Majorana-assisted teleportation”, to distinguish it from non-local transport considered by us, where quasi-Majoranas can be potentially involved.

In summary I am happy to recommend the manuscript for publication as long as the authors can modify the manuscript to clarify the concerns.

We gratefully thank Referee B again, for the recommendation to publish our work.

List of changes

In the main text:

1. We change “topological superconducting island” in the title to “**Majorana-hosted** superconducting island”.
2. We change “Indeed, the non-local transport through a topological SC island [31, 32], known as the Majorana-assisted teleportation (MT)...” in page 1 to “Indeed, the non-local transport through a topological SC island [31, 32], known as the **topological** Majorana-assisted teleportation (MT)...”.

3. In Fig.1, we change the typeface of the “Log” and the “Max” to Times new Roman. Change the typeface of the “E” to italic.
4. We get rid of the italic bold of the “process A”, “process B”, and “coherent double MT”.
5. We change “these two processes decohere, and only single coherent MT exists.” in the caption of Fig.2 to “**these two processes become incoherent.**”. Change “ Γ_{seq} ” to “ **Γ_{R}** ”.
6. In Fig.2, we change the “coherent double Fu-teleportation” to “**Coherent double Majorana-assisted teleportation**”, change “single coherent Fu-teleportation” to “**Incoherent double Majorana-assisted teleportation**”, change “(i)” and “(ii)” to “**(a)**” and “**(b)**”, change the color of the leads to yellow.
7. In Fig.3, we change all the equations to italic.

In the supplementary information

8. We change “Supplementary material” in the title to “**Supplementary Information**”.
9. We add “**Supplementary**” in front of Eq. ()/Fig. when quoting the equations/figures in the SI.
10. We change Eq. (10) in page 5 to “**Supplementary Eq. (16)**”.
11. In Supplementary Fig.1, we change the typeface of the “G” and “T” to italic.
12. We change “combining Eq. (7) and Eq. (8) in the main text” to “combining Eq. (7) and Eq. (8) of the main text”.